# Glial Cultures Differentiated from iPSCs of Patients with *PARK2*-Associated Parkinson’s Disease Demonstrate a Pro-Inflammatory Shift and Reduced Response to TNFα Stimulation

**DOI:** 10.3390/ijms24032000

**Published:** 2023-01-19

**Authors:** Tatiana Gerasimova, Ekaterina Stepanenko, Lyudmila Novosadova, Elena Arsenyeva, Darya Shimchenko, Vyacheslav Tarantul, Igor Grivennikov, Valentina Nenasheva, Ekaterina Novosadova

**Affiliations:** 1Laboratory of Cell Differentiation, Institute of Molecular Genetics of National Research Centre “Kurchatov Institute”, Moscow 123182, Russia; 2Laboratory of Molecular Neurogenetics and Innate Immunity, Institute of Molecular Genetics of National Research Centre “Kurchatov Institute”, Moscow 123182, Russia

**Keywords:** Parkinson’s disease, *PARK2* mutations, iPSC-derived glial cells, neuroinflammation, TNFα stimulation

## Abstract

Parkinson’s disease (PD) is the second most common neurodegenerative diseases characterized by progressive loss of midbrain dopaminergic neurons in the substantia nigra. Mutations in the *PARK2* gene are a frequent cause of familial forms of PD. Sustained chronic neuroinflammation in the central nervous system makes a significant contribution to neurodegeneration events. In response to inflammatory factors produced by activated microglia, astrocytes change their transcriptional programs and secretion profiles, thus acting as immunocompetent cells. Here, we investigated iPSC-derived glial cell cultures obtained from healthy donors (HD) and from PD patients with *PARK2* mutations in resting state and upon stimulation by TNFα. The non-stimulated glia of PD patients demonstrated higher *IL1B* and *IL6* expression levels and increased IL6 protein synthesis, while *BDNF* and *GDNF* expression was down-regulated when compared to that of the glial cells of HDs. In the presence of TNFα, all of the glial cultures displayed a multiplied expression of genes encoding inflammatory cytokines: *TNFA*, *IL1B*, and *IL6*, as well as IL6 protein synthesis, although PD glia responded to TNFα stimulation less strongly than HD glia. Our results demonstrated a pro-inflammatory shift, a suppression of the neuroprotective gene program, and some depletion of reactivity to TNFα in *PARK2*-deficient glia compared to glial cells of HDs.

## 1. Introduction

Parkinson’s disease is one of the most common neurodegenerative disorders that affects people of different age categories and significantly reduces quality of life, since it is accompanied by severe neurological manifestations. Disease pathogenesis is characterized by two main features: progressive loss of dopaminergic neurons in the substantia nigra pars compacta (SN) and, in some cases, intracellular accumulation of misfolded α-synuclein in Lewy bodies and Lewy neurites [1,2,3]. Neuroinflammation is also a significant pathogenetic factor of PD [4,5,6]. Supporting evidence can be found in literature data, confirming that serum and brain tissue levels of inflammatory cytokines are elevated in PD patients compared to healthy controls. Notably, increased serum TNFα levels have been found to be positively correlated with PD clinical characteristics [7]. A comprehensive study of protein profile of human SN and striatum samples from healthy donors (HD) and PD patients has shown that the factors that were downregulated were either inflammatory antagonists or growth factors, while the factors that were upregulated were inflammatory cytokines, including TNFα [8].

Sustained neuroinflammation is regulated through complex communications between neuroglia on the one hand, and peripheral immune cells infiltrating central nervous system (CNS) in pathological conditions on the other hand. While the immune system in the CNS is represented by microglia—resident brain macrophages, other glial cell populations, namely astrocytes and oligodendrocytes, also play an important role in chronic neuroinflammation in PD [6,9,10].

The normal astrocyte function is to protect neurons by releasing neurotrophic factors [11,12] and antioxidants [13], and by disposing of microglial cellular debris [14]. In PD, astrocytes play a dual role: they can either support dopaminergic neuron function and regeneration or cause neuronal cell death [15]. The neuroprotective function of astrocytes is well-established and includes the cleaning of extracellular α-synuclein aggregates, the activation of the Nrf2 signaling pathway with subsequent producing various antioxidative molecules, and even the donation of mitochondria to injured dopaminergic neurons [16]. At the same time, being immunocompetent, astrocytes respond to inflammatory ligands, exhibit a pro-inflammatory gene expression profile, and produce pro-inflammatory cytokines [16,17]. When exposed to various different stimuli, astrocytes acquire multiple distinct activation phenotypes depending on the local microenvironment [18]. Nevertheless, two opposite functional states of activated astrocytes are generally distinguished: the neurotoxic A1 type that is dominant in inflammation-induced lesions and the neuroprotective A2 type [19].

It is known that in PD patients activated microglia release IL1α, TNFα, and C1q that, in their turn, induce neurotoxic A1 type astrocytes [15,20]. Besides, as shown in the α-synuclein preformed fibrils (PFF) PD model, astrocyte cultures acquire a pronounced pro-inflammatory secretory profile even without the influence of microglial factors [21]. It has also been reported that astrocytes with accumulated α-synuclein produce proinflammatory cytokines, including IL1β and TNFα, as well as chemokines, such as CXCL1 and CX3CL1 [2]. The increased production of pro-inflammatory molecules via reactive glia is detrimental to neurons and causes neurodegeneration [5,22]. As a result of disruption of the blood-brain barrier (BBB), pro-inflammatory cytokines and peripheral immune system cells from peripheral blood enter brain tissues. These cells are an additional source of inflammatory mediators that further contribute to the activation of astrocytes. Thus, glial cells in PD are constantly exposed to an inflammatory environment. In response, they initiate distinct transcriptional programs, including those of the inflammatory kind and those aimed at compensatory synthesis of regulatory cytokines and neurotrophic factors.

Several studies have determined that the main PD-associated genes are expressed in astrocytes. Mutations in one of them, *PARK2*, are associated with early onset familial form of PD [16]. Parkin, encoding using the *PARK2* gene, is involved in astrocyte-specific functions, including an inflammatory response and the synthesis of neurotrophic factors [5].

Astrocytes derived from induced pluripotent stem cells (iPSCs) represent a novel and promising tool for studying patient-specific human neuroglia that provides the opportunity to obtain genetically identical cells in a scalable way. Glial cultures differentiated from human iPSCs are useful for modeling neurological diseases with an inflammatory component and for investigation of the role of astrocyte dysfunction in neurodegeneration [18,23,24,25].

The aim of our study was to model inflammatory conditions in vitro and evaluate the differences in reactivity between iPSC-derived glial cultures obtained from HDs and those differentiated from PD patients with mutations in the *PARK2* gene.

## 2. Results

### 2.1. HD and PD Glial Cultures Contained High Percentage of Astrocytes

Glial cultures were obtained from iPSCs of two HD and two PD patients with different mutations in the *PARK2* gene according to the protocol of directed differentiation [26]. The cell cultures were shown to contain 95.5 ± 0.74% S100-positive cells with no statistical difference between each other (Figure 1).

### 2.2. Cytokine Treatment didn’t Affect Cell Proliferation in Glial Cultures

We estimated the growth dynamics of non-stimulated and stimulated (with TNFα or TNFα + IL1β) HD and PD glial populations using MTT method at three time points: (1) after 24 h, (2) after 72 h of cultivation, (3) after 72 h of cultivation, the cultures were washed twice with sterile DPBS, and then fresh growth medium was added for the next 72 h of cultivation (72hw72h). No significant difference in cell count was found between non-stimulated and stimulated cells within each cell culture at all the time points (Figure 2). Evidently, cell growth was a characteristic feature of each individual cell line. We chose not to use inhibitors of cell proliferation so as not to provoke changes in cell transcriptome and metabolic activity. Instead, for data normalization, IL-6 concentrations were recalculated as pg/mL per 10^3^ cells.

In our preliminary experiments, we found that iPSC-derived glial cultures only slightly increase IL6 production in response to LPS stimulation. The above finding was consistent with recent study that examined the respective abilities of astrocytes and microglia in vitro and proved that microglia is a major glial cell type that induces TNFα, IL1β, and IL6 production when stimulated by LPS [27]. In another work, extended co-stimulation of human iPSC-derived astrocyte cultures with microglia-derived factors, IL1β and TNFα, produced a reactive astrocyte phenotype with a broad inflammatory secretion profile [18].

Taking into account the differentiation protocol used in our study, the glial cultures did not include microglia, as a derivative of another germ layer. Therefore, treatment with TNFα and IL1β modelled the influence of factors produced by inflammatory microglia on glial cells. As a criterion to estimate glial reactivity, we chose the degree of change in the following parameters: expression of genes encoding the main inflammatory cytokines (*TNFA*, *IL1B* and *IL6*), and neurotrophic factors (*BDNF*, *GDNF*), as well as IL6 protein synthesis.

### 2.3. Expression of Genes Encoding Pro-Inflammatory Cytokines and Neurotrophic Factors Differed between iPSC-Derived Glia from HD and Patients with PARK2-Associated Parkinson Disease (PD) in Both Steady State and Inflammatory Conditions

Using the qPCR method, we studied the expression of genes encoding the main inflammatory cytokines *TNFA*, *IL1B* and *IL6*, genes of neurotrophic factors *BDNF*, *GDNF*, and the *SNCA* gene that is closely associated with PD pathogenesis. The close relationship between the chosen genes is demonstrated in Figure 3.

We observed that in resting PD glia, gene encoding neurotrophic factor *BDNF* was expressed at a lower level than in HD cells (2.6-fold), while the expression level of the *GDNF* gene didn’t differ significantly. The expression of *TNFA* differs somewhat between cultures, but without a clear direction. The expression of other pro-inflammatory genes, *IL6* and *IL1B*, was significantly higher in PD glial cells in comparison to HD ones (2.4-fold and 4.1-fold, respectively) (Figure 4a).

In the presence of TNFα cells multiplied expression of genes encoding inflammatory cytokines: *TNFA*, *IL1B*, and *IL6*. The peak fold change of *TNFA* expression in both groups was reached after 24 h of cultivation (9.0-fold in HD and 3.45-fold in PD) with a subsequent decrease at 72 h (5.6-fold in HD and 1.3-fold in PD). Maximum expression of *IL1B* and *IL6* in PD was also observed at 24 h (18.5-fold and 8.0-fold, respectively) and then the expression of both genes decreased from 24 h to 72 h (6.9-fold for *IL1B* and 1.6-fold for *IL6*). The opposite dynamics was found for *IL1B* and *IL6* genes in HD: expression increased from 24 h to 72 h, and the peak was reached at 72 h (27.6-fold for *IL1B* and 19.4-fold for *IL6*).

We observed some other differences between the glial cultures of HD and PD patients. In PDs, *IL1B* and *IL6* up-regulation levels were lower than in HDs at 72 h time point (4.0-fold and 12.4-fold, respectively). Washing adhesive cell cultures with DPBS and changing the medium to a non-inflammatory one after stimulation with TNFα sharply reduced the expression of genes encoding inflammatory cytokines in all the glial populations although in HDs it remained higher than the basal levels.

At 24h, *BDNF* expression did not demonstrate any significant changes neither in HD nor in PD. After 72 h of stimulation *BDNF* was significantly up-regulated in HD, while in PD it was down-regulated. *GDNF* expression, on the contrary, was elevated several times, with no statistical difference between HD and PD cells at 24 h and 72 h. After a DPBS wash, *BDNF* expression decreased in PD to half of its initial level, while returning to base level in HD. At the same time, *GDNF* expression remained elevated in the HD cells (by 8.6-fold increase), while returning to baseline in the PD cells (Figure 4b).

### 2.4. SNCA Gene Expression Was Stable

*SNCA* gene expression didn’t demonstrate significant change during the time of cultivation, didn’t differ between HD and PD glial cells, and was not affected by stimulation with TNFα in both HD and PD cultures. Surprisingly, washing with DPBS upregulated *SNCA* expression level in stimulated HD cultures and showed the same tendency in non-stimulated HD cells, although statistical significance was not reached. In the PD cultures *SNCA* expression remained stable (Figure 5).

### 2.5. IL6 Protein Secretion Was Upregulated in Resting Glial Cultures of PD Patients with PARK2 Gene Mutations Compared to HD Cells

Using ELISA, we found the higher basal level of IL6 secretion in glial cultures differentiated from iPSCs of PD patients with *PARK2* mutations when compared to those from HDs at 24 h and 72 h time points (4.5-fold and 4.8-fold, respectively). Washing with DPBS was obviously stressful for HD populations and was accompanied by an increase in IL6 synthesis (Figure 6a).

Treatment with TNFα alone increased production of IL6 in all the studied glial populations as detected by ELISA. Significant differences in the strength of cell activation were revealed between HD and PD glial cultures in both variants of stimulation. HD glia responded with a greater FC increase in IL6 secretion compared to PD cells at 24 h (8.6-fold and 2.4-fold increase) and 72 h (9.3-fold and 3.1-fold increase). After washing stimulated cells with DPBS all the glial populations down-regulated IL6 production, although in HDs it remained higher than the basal levels (Figure 6b). Combined treatment with two inflammatory cytokines (TNFα + IL1β) had the same effect, but more strong and more rapid, since the maximum secretion occurred within 24 h (175.1-fold and 17.3-fold increase) and decreased up to 72 h (69.5-fold and 9.1-fold increase). The differences between HD and PD cultures in the presence of TNFα + IL1β demonstrated the same direction as found in case of TNFα (Figure 6c). Thus, at protein level IL6 increased at 24 h and 72 h of inflammatory stimulation in HD and PD glial cultures, although, HD cells showed significantly greater degree of up-regulation than PD. Notably, another shift was observed in IL6 gene expression. Maximum IL6 expression in PD was observed at 24 h (8.0-fold increase) and then the gene expression decreased from 24 h to 72 h (to 1.6-fold increase), while in HD the peak was reached at 72 h (19.4-fold increase) (Figure 4b).

Overall, our observations may indicate that, in steady state, inflammatory gene expression (*IL1B*, *IL6*) and protein synthesis (IL6) were up-regulated, while the neuroprotective gene expression (*BDNF*, *GDNF*) was suppressed in glia of PD patients with mutations in the *PARK2* gene compared to HD cells. Stimulation with TNFα activated both the pro-inflammatory and the neuroprotective gene network in all the glial cultures. Based on the data obtained, it could be assumed that, in response to inflammatory milieu, HD glia up-regulated the pro-inflammatory cytokine program to a greater extent than PD cells, and compensatory neuroprotective gene activation in HD glial cells lasted longer than in PD glia. The *SNCA* gene expression had no difference between HD and PD glial cultures and was not affected by stimulation with TNFα.

## 3. Discussion

It is known that astrocytes and oligodendrocytes are the most numerous glial cell subpopulations in the human brain and play a significant role in PD, although, to date, the immunological aspect of astrocytes has been studied in more detail compared to oligodendrocytes. Recent evidence indicates that neuroinflammation is generally pro-resolving in the acute phase, but after chronicization it promotes neurodegeneration [28,29]. Therefore, the ability of astrocytes and oligodendrocytes to produce immunoregulatory cytokines and neurotrophic factors in response to the inflammatory challenge is very important for the outcome of the neuropathology. An increased number of reactive astrocytes in the SN of patients with PD was recently reported [11]. The goal of this study was to compare immunocompetent resource of glial cells differentiated from iPSCs of HDs to those obtained from PD patients with *PARK2* mutations, both in a steady state and under inflammatory conditions.

Treatment with TNFα alone or with a combination of TNFα and IL1β in this study modelled the influence of factors produced by inflammatory microglia on astrocyte cultures. TNFα is produced by microglia, i.e., activated resident macrophages of the brain, and is one of the major mediators of neuroinflammation in neurodegenerative disorders [30,31]. TNFα has a dual influence of neurotoxicity and neuroprotection in the brain since it acts through two receptors, TNFR1 and TNFR2 [32], which are both expressed in astrocytes [33]. Interestingly, glial cells not only react to TNFα, but also produce it into a local environment. In the recent study, iPSC-derived astrocytes stimulated by TNFα acquired a typical phenotype of astrogliosis characterized by nuclear translocation of NF-kB, production of cytokines, and changes in morphology and function [24,34]. IL1β is released from microglia after a neural injury and is one of the most potent signals for activating astrocytes. Astrocytes express AP-1, NF-κB, and several different mitogen-activated protein kinases (MAPKs), which are the key elements of the transcriptional program of cell response to increased IL1β level [35]. It is proposed that sustained but not acute increase in IL-1β expression, has toxic effects on SN neurons [4].

We estimated the expression of genes encoding the main inflammatory cytokines (*TNFA*, *IL1B* and *IL6*), and neurotrophic factors (*BDNF*, *GDNF*), as well as the level of IL6 protein synthesis, since the importance of these participants of neuroinflammation is well established.

IL6 is considered a major cytokine in the central nervous system [36]. Showing pronounced inflammatory properties, IL6 acts as immunomodulatory factor at the same time, and may have a bi-directional influence on neuron survival and function [37,38]. These properties are supported by other findings: IL6 up-regulation in cerebrospinal fluid of PD patients reversely correlates with the severity of clinical symptoms [36]; in PD, IL6 reduces the cytotoxicity of Ca2+ and ROS [36], plays a neurotrophic role for midbrain dopaminergic neurons [38], and promotes survival of oligodendrocytes [39]. Moreover, during PD, astrocyte-derived IL6 in cooperation with sIL-6R, which is secreted by apoptotic neurons, mediates adult neurogenesis, i.e., the differentiation of new neurons and glia from neural stem cells [38,40,41].

BDNF mRNA levels in human astrocyte cultures were found to be very low under normal conditions. Similarly, in human brain samples, only oligodendrocytes showed a very low amount of GDNF mRNA. In contrast, in neurodegenerative disease, elevated BDNF mRNA was observed in astrocytes surrounding amyloid plaques, while GDNF mRNA was elevated in astrocytes of lesioned striatum in the 6-hydroxydopamine (6-OHDA) model of PD [12]. Astrocytes receive signals from damaged neuronal cells and produce GDNF that has a direct effect on promoting survival and morphological differentiation of certain populations of neurons, including the dopaminergic nigrostriatal pathway [16,42,43].

According to our results, in resting state, iPSC-derived glial cultures of PD patients with the *PARK2* mutations demonstrated significantly higher expression of *IL1B*, *IL6* and lower expression of *BDNF* genes, and secreted IL6 protein more actively compared to glia obtained from HD. Our observations may point to a slight shift to inflammatory status, on the one hand, and to suppression of compensatory mechanisms, on the other hand, in PD glial cells compared to control even in a non-inflammatory milieu. Stimulation of all the glial cell lines by TNFα caused increased expression levels of genes not only encoding pro-inflammatory cytokines, *TNFA*, *IL1B*, *IL6*, and neuroprotective *GDNF*, as well. It also led to up-regulated secretion of IL6 protein. All these events were more pronounced in HDs, than in PDs glia cells, showing less reactivity potential of diseased cells. The similar, but more rapid and strong glial reactivity, was observed when the cell cultures were co-stimulated by TNFα and IL1β, which is in consistence with recent observations of synergistic effect of these two cytokines [44]. Inflammatory environment forced glial cells to increase by many fold the expression of *GDNF*, but not *BDNF*. Interestingly, after removing inflammatory medium and PBS wash, *BDNF* and *GDNF* expression level remained elevated in HDs by 2.3- and 8.6-fold, respectively, while returned to baseline in PD patients glia, showing statistically significant differences between healthy and diseased cells. We assumed that *GDNF* up-regulation, as a part of neuroprotective system, could be more effective in HDs than in PDs.

We supposed that the differences in reactivity might be associated with change of Parkin function in glia with different *PARK2* mutations. It is known that not only dopaminergic neurons are enriched with Parkin, but astrocytes are, as well [6,45]. In a recent work, a hypothesis was put forward that Parkin influences inflammation in PD through its effect on the integrity or quality of mitochondria, rather than acting directly [29]. Parkin marks dysfunctional mitochondria for degradation through mitophagy and maintains cellular respiration. Consequently, the lack of this protein might lead to a decrease in astrocyte proliferation and altered secretory activity [15,34]. Nevertheless, mutation in the *PARK2* gene may directly influence on the expression of *TNFA*, *IL1B*, *IL6*, *BDNF*, and *GDNF* via several pathways related to inflammatory and neuroprotective mechanisms. Astrocytes deficient in Parkin display up-regulation of the nucleotide-oligomerization domain receptor 2 (NOD2) that was identified as a Parkin substrate for ubiquitylation and degradation. The proteasomal degradation of NOD2 is critical for the maintenance of normal astrocyte neurotrophic function [45]. That is consistent with significantly reduced *BDNF* and *GDNF* expression demonstrated in Parkin KO astrocytes [45,46]. However, with regard to activation of inflammatory mechanisms in glia with *PARK2* mutations, the literature data remain scarce and controversial. Wang Y. et al. reported that upon TNFα stimulation, Parkin ubiquitinylates RIPK1, which promotes activation of nuclear factor-κB (NF-κB) with p65 nuclear translocation, and MAPKs. Parkin also promotes recruitment of the transforming growth factor β (TGF-β)-activated kinase 1 (TAK1), NF-κB essential molecule (NEMO), Sharpin, and A20. All the aforementioned factors are associated with inflammatory TNFR1 signaling [47]. In other work, pathogenic Parkin mutants with impaired neuroprotective capacity also showed the reduced ability to stimulate NF-κB-dependent transcription [48]. On the contrary, it has been reported that Parkin-knockout astrocytes exhibit increased ER stress and cytokine production [45]. This discrepancy could possibly be explained by the fact that primary astrocytes used in this study could contain an admixture of microglial cells. Another work demonstrated that brain inflammation in Parkin-deficient mice was comparable to wildtype mice [49].

Our results regarding glial cultures from patients with mutations in *PARK2* seems to have differences with some evidence concerning the reactivity of glia in PD associated with mutations in other genes. Recent study in astrocytes derived from iPSCs of *LRRK2 G2019S* mutant patients, with one patient also carrying a *GBA N370S* mutation, in comparison to healthy cells, confirmed the up-regulated expression of α-synuclein and the increased release of cytokines upon inflammatory stimulation [50]. *PINK1* loss increased expression of IL1β and TNFα in inflammatory stimulated astrocytes, with no change in IL6 expression [51]. A multiplex immunoassay in the conditioned media from naïve *GBA* mutant astrocytes evaluated significantly decreased basal levels of proinflammatory cytokines, including IL6, IL1β, and TNFα, when compared with control. On the contrary, LPS-triggered cytokine expression was dramatically reduced by *GBA1* mutation cells [52], which is similar to our results, showing a robust deficit in upstream transcriptional programs of inflammatory response.

*SNCA* is abundantly expressed in neurons in PD, whereas in astrocytes *SNCA* expression is low [2,5]. Nevertheless, the protein has functional relevance given its important physiological roles affecting synaptic transmission, dopamine release, microglial function, and membrane homeostasis [53]. PD is often accompanied by α-synuclein pathology. Recent study demonstrated that the mRNA expression level of the *SNCA* gene was increased in PD astrocytes differentiated from iPSCs of *LRRK2* and *GBA* mutation at all time points during the four-month cultivation [50]. Whereas α-synuclein accumulation is also found in virtually all cases of sporadic PD, in *PARK2*-associated PD, α-synuclein may be a minor player [3]. Our findings were consistent with these observations; since *SNCA* expression didn’t show significant differences between HD and PD glial cultures, it was stable during the time of cultivation, and was not affected by stimulation.

Interestingly, in our study, *SNCA* expression was up-regulated in healthy glial cells after washing with DPBS, both in resting and stimulated cells. We assumed, that such an effect was associated with the well-established influence of PBS on cell adhesion on substrate that decreases if PBS lacks calcium and magnesium [54]. These observations were supported by Wersinger, C., et al., which demonstrated *SNCA* low expression at focal adhesion points [55]. The lack of *SNCA* is associated not only with cell adhesion, but with apoptosis, as well [56], so that *SNCA* up-regulation after PBS washing could be compensatory reaction that occurs in HD cells but not in PD cells. Notably, secretion of IL6, protein with modulatory properties, increased many folds after PBS washing in HD glia, while in PD cultures it didn’t change significantly.

Taken together, we assumed that glia in *PARK2*-associated PD had more inflammatory status in the resting state. The pro-inflammatory shift was accompanied by suppression of compensatory neuroprotective mechanisms. At the same time, *PARK2-*deficient glia from PD patients responded less strongly than HD glia to inflammatory challenge in the local microenvironment, possibly demonstrating some depletion of activation capacity.

To our knowledge, in this study for the first time the differences in reactivity to TNFα between iPSC-derived glial cultures from HD and from PD patients with different mutations in *PARK2* gene were revealed. Additional comparative studies of glial cultures derived from PD patients with mutations in other genes will help to identify common and specific reactivity patterns.

## 4. Materials and Methods

### 4.1. Ethics Statement

The study complies with the World Medical Assembly Declaration of Helsinki–Ethical Principles for Medical Research Involving Human Subjects. This work was approved by the Ethic Committee of the Institute of Molecular Genetics of National Research Centre “Kurchatov Institute” (protocol №3 from 19 February 2018). Every PD patient and healthy donor provided a written informed consent.

### 4.2. Generation of iPSCs

iPSC lines of two HDs (HD1, HD2) and one PD patient with het *EX2 dup PARK2* (PD1) were previously generated [26,57]. iPSCs of another PD patient with *PARK2* mutation (PD2) were obtained as described below.

Skin fibroblast cells from the patient with het *EX2 del PARK2* were reprogrammed using the CytoTune™-iPS 2.0 Sendai Reprogramming Kit (Thermo Fisher Scientific, Waltham, MA, USA) according to manufacturer’s instructions. The cells were transduced with the Sendai reprogramming vectors at the MOI 5 (each vector). Infected cells were transferred with 0.05% trypsin on Matrigel (Corning, NY, USA) 1:5 by area. Then, the cells were cultured in human pluripotent cell culture medium (DMEM/F12 (Gibco, Billings, MT, USA), 20% KO SR (Invitrogen, Carlsbad, CA, USA), 2 mM L-glutamine (ICN Biomedicals Inc., Costa Mesa, CA, USA), 0.1 mM β-mercaptoethanol (Sigma-Aldrich, Saint Louis, MO, USA), 1% non-essential amino acids, bFGF 10 ng/mL, penicillin-streptomycin (50 U/mL; 50 μg/mL) (all Paneco, Moscow, RF). The medium was changed every day. On day 15 after infection, the selection of clones was started, based on the morphological criteria of similarity of colonies with human ESCs colonies. After 2–4 passages, the iPSCs were transferred to the mTeSR1 (Stem Cell Technologies, Vancouver, British Columbia, Canada) medium on a Matrigel substrate (Corning, NY, USA) in CO_2_-incubator (5% CO_2_, 80% humidity and 37 °C)

### 4.3. Embryoid Bodies Formation

For the formation of embryoid bodies, the iPSCs were removed from the substrate and transferred to Ultra low adhesion plates (Corning, NY, USA) and cultured in DMEM/F12 (Gibco, Billings, MT, USA), 20% KO SR (Invitrogen, Carlsbad, CA, USA) 5% FBS (Hyclone, Logan, UT, USA), 0.1 mM β-mercaptoethanol (Sigma-Aldrich, Saint Louis, MO, USA), 1% NEAA (Hyclone, Logan, Utah, USA), penicillin-streptomycin (50 U/mL; 50 µg/mL) (Paneco, Moscow, RF). Medium was changed every 2 days. Embryoid bodies were grown for 10 days, then plated to gelatin-coated Petri dishes (Corning, NY, USA) and cultured for 21 days in the same medium. Differentiated cells were fixed with 4% PFA, stained with antibodies to specific markers, and investigated with an AxioImager Z1 fluorescence microscope equipped with an AxioCam HRM camera using AxioVision 4.8 software (Zeiss, Oberhohen, Germany) (Figure 7).

### 4.4. Obtaining of Glial Cell Lines

The differentiation of iPSCs in glial direction was performed as described earlier [58]. The glial progenitors passed the full course of differentiation according to the protocol and were stained with astrocyte specific marker S100. Table 1 presents a description of glial cell lines used in the study.

### 4.5. Immunocytochemistry

The adherent cells on the Petri dish were washed with PBS, fixed with 4% paraformaldehyde in PBS (pH 6.8) for 20 min at room temperature (RT), washed in PBS with 0.1% Tween 20 (Sigma-Aldrich, Saint Louis, MO, USA) three times for 5 min. Nonspecific antibody sorption was blocked by incubation in blocking buffer (PBS with 0.1% Triton x100 and 5% fetal bovine serum (HyClone, Logan, UT, USA)) for 30 min at RT. Primary antibodies (Table 2) were applied overnight at 4 °C, and then washed in PBS with 0.1% Tween 20 three times for 5 min. The secondary antibodies were applied for 60 min at RT, then washed in PBS with 0.1% Tween 20 three times for 5 min. After that, the cell cultures were incubated with 0.1 μg/mL DAPI (Sigma-Aldrich, Saint Louis, MO, USA) in PBS for 10 min for visualization of the cell nuclei and washed twice with PBS. The cells were investigated using the AxioImager Z1 fluorescence microscope (Carl Zeiss, Oberhohen, Germany), and images were taken with AxioVision 4.8 software (Carl Zeiss, Oberhohen, Germany). For cell counting, the multiple fields that cover the whole dish surface were imaged. The obtained images were analyzed with ImageJ 1.49 software (NCBI, Bethesda, MD, USA) using ITCN plugin (Center for Bio-image Informatics, Santa Barbara, CA, USA).

### 4.6. Inflammatory Stimulation of Glial Cell Cultures

Differentiated glial populations were dissociated with trypsin and seeded on a 96-well plate with 5 × 10^3^ cells/well in 200 µL of growth medium for glial progenitors (DMEM/F12 (Gibco, Billings, MT, USA) containing 1 mM non-essential amino acids (Paneco, Moscow, RF), 2 mM L-glutamine (ICN Biomedicals Inc., Costa Mesa, CA, USA), penicillin-streptomycin (50 U/mL; 50 µg/mL) (Paneco, Moscow, Russia), 1% N2 supplement (Life Technologies, Carlsbad, CA, USA), B-27 supplement (Life Technologies, Carlsbad, CA, USA), 8 ng/mL FGF2, 10 ng/mL Heregulin, 200 ng/mL IGF1, 10 ng/mL Activin A (all STEMCELL Technologies, Vancouver, BC, Canada)) that contained Rock-inhibitor (STEMCELL Technologies, Vancouver, British Columbia, Canada). The next day, the cells were cultured in one of the following ways: (1) No stim—in fresh growth medium for 24 h and 72 h; (2) TNFα-Stim—in medium containing 10 ng/mL TNFα (#PSG250-10, SCI-store, Moscow, RF) for 24 h and 72 h; (3) 72hw72h-No stim and 72hw72h-Stim—after 72 h of cultivation with/without stimuli, glial cultures were washed twice with sterile DPBS (Paneco, Moscow, RF), and fresh growth medium was added for the next 72 h of cultivation; (4) TNFα + IL1β-Stim—in the presence of 10 ng/mL TNFα and 10 ng/mL IL1β (#I-9401, Sigma, Saint Louis, MO, USA) for 24 h and 72 h. The main subject of the study was the stimulation by TNFα, while the combination TNFα + IL1β was used to estimate the synergistic effect of these two main proinflammatory cytokines. The above groups will be referred to by their notations throughout this work.

### 4.7. MTT (3-(4,5-Dimethylthiazol-2-yl)-2,5-diphenyltetrazolium bromide) Assay

For quantification of cell growth and viability, adhesive cell cultures were incubated for 4 h in culture medium containing 0.5 mg/mL MTT (Sigma-Aldrich, Saint Louis, MO, USA). Next, the medium was removed and blue MTT–formazan product was diluted with DMSO (Panreac, Barselona, Spain). After 2 h of incubation at RT on a shaker setting of 150 rpm/min in the dark, absorbance of the formazan solution was recorded at 600 nm using spectrophotometer (Metertech, Nankang, Taipei, Taiwan). Statistical analysis of the data was performed using two-way ANOVA with multiple comparisons corrected with Bonferroni test (GraphPad Prism 8.0.1. software). Results were presented as mean ± SEM.

### 4.8. ELISA

The concentration of IL6 protein in culture medium was estimated using a human IL-6 ELISA kit (Vector-Best, Moscow, RF) according to the manufacturer’s protocol. Absorbance was measured at 450 nm (Metertech, Nankang, Taipei, Taiwan). The concentrations obtained were recalculated per cell count in each well and were used in the analysis as concentrations per 1 × 10^3^ cells. The fold change (FC) of IL6 concentration was calculated relative to mean value in non-stimulated cells at each time point. Statistical analysis of the data was performed using two-way ANOVA with multiple comparisons corrected with Bonferroni test (GraphPad Prism 8.0.1. software). The results were presented as mean ± SEM (at least three experiments). The differences were considered statistically significant at * *p* < 0.05, ** *p* < 0.05, *** *p* < 0.005, **** *p*< 0.0001.

### 4.9. Quantitative Real-Time PCR (qPCR)

Total RNA was extracted from the cells with a Trizol RNA purification kit (Invitrogen, Carlsbad, CA, USA) following the manufacturer’s instructions, with subsequent DNA-free DNA Removal Kit (Invitrogen, Carlsbad, CA, USA) treatment. cDNA was synthesized on 0.5–2 μg of total RNA using M-MLV Reverse Transcriptase (Evrogen, Moscow, RF) with random primers. The primer sequences are shown in Table 3. The cDNA obtained was amplified using LightCycler 96 instrument (Roche, Basel, Switzerland) set to the following reaction conditions: denaturation (preincubation) at 95 °C (5 min), amplification cycles n = 45 (94 °C, 20 s; 60–69 °C, 20 s; 72 °C, 20 s), melting (95 °C, 10 s; 65 °C, 60 s; 97 °C, 1 s), cooling at 37 °C, 30 s. qPCRmix-HS SYBR reaction mixture (Evrogen, Moscow, RF) was used. *18S rRNA* was accepted as a reference gene. Changes in expression levels of target genes were calculated using the 2^∆∆Ct^ method. The fold change (FC) of expression was calculated relative to mean value in non-stimulated cells at each time point. Statistical analysis of the qPCR data was performed using unpaired two-tailed t-test and two-way ANOVA with multiple comparisons corrected with Bonferroni test (GraphPad Prism 8.0.1. software). The results were presented as mean ± SEM (n = 3). The differences were considered statistically significant at * *p* < 0.05, ** *p* < 0.005, *** *p* < 0.005, **** *p* < 0.0001.

## Figures and Tables

**Figure 1 ijms-24-02000-f001:**
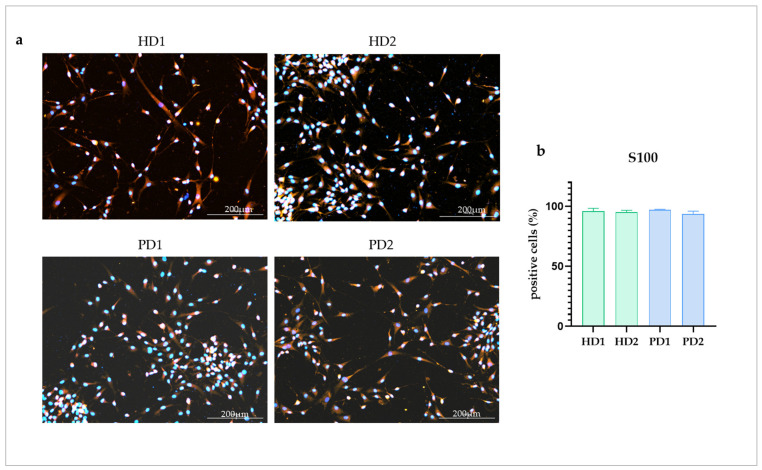
Immunocytochemical analysis of astrocyte marker S100 in the glial cell lines. (**a**), images from glial cultures obtained from healthy donors (HD1, HD2) and from patients with *PARK2*-associated PD (PD1, PD2). Red—S100, blue—DAPI. (**b**), percentage of S100 positive cells in the studied glial cultures. The data are shown as mean ± SEM.

**Figure 2 ijms-24-02000-f002:**
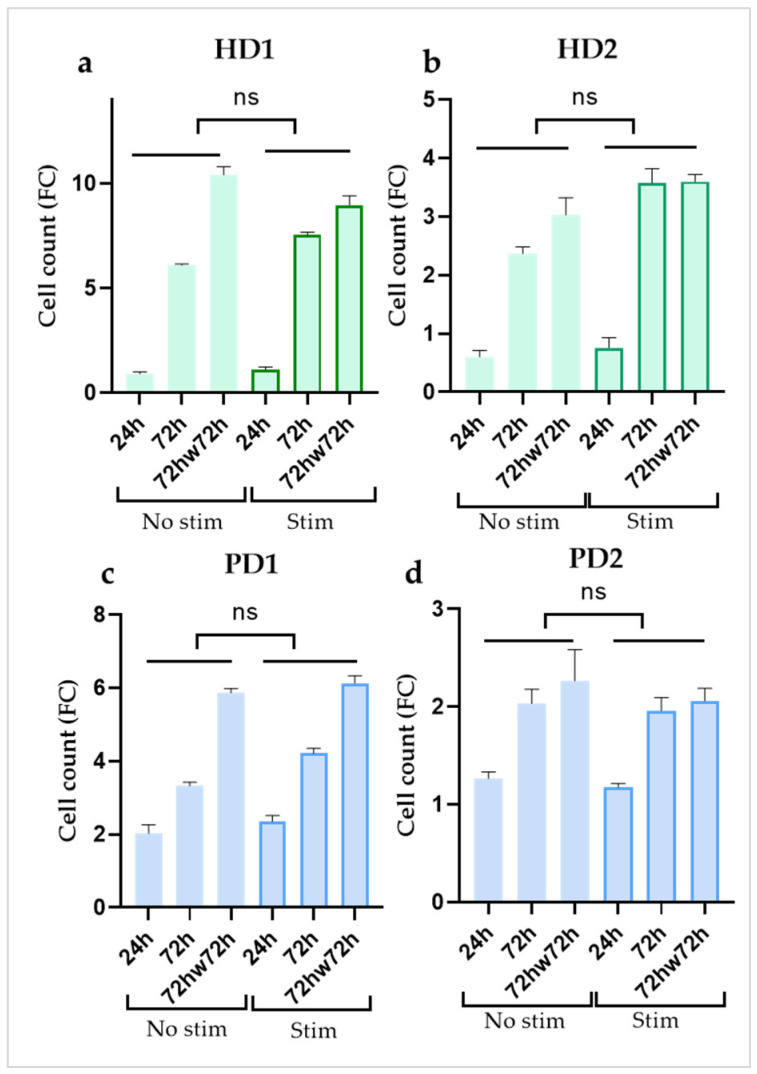
Comparison of the dynamics of cell growth in the absence or presence of inflammatory cytokines (**a**,**b**) in glial cultures from healthy donors (HD1, HD2); (**c**,**d**) in glial cultures from PD patients with *PARK2* mutations (PD1, PD2) during cultivation. Cell counts were estimated using MTT test. Fold change (FC) in the cell count in each well (n = 3) was calculated relative to the initially seeded 5 × 10^3^ cells per well. The data are shown as mean ± SEM at three time points (24 h, 72 h, 72hw72h) for each glial culture. Two-way ANOVA and multiple comparisons corrected with Bonferroni test were used for statistical processing; ns, non-significant.

**Figure 3 ijms-24-02000-f003:**
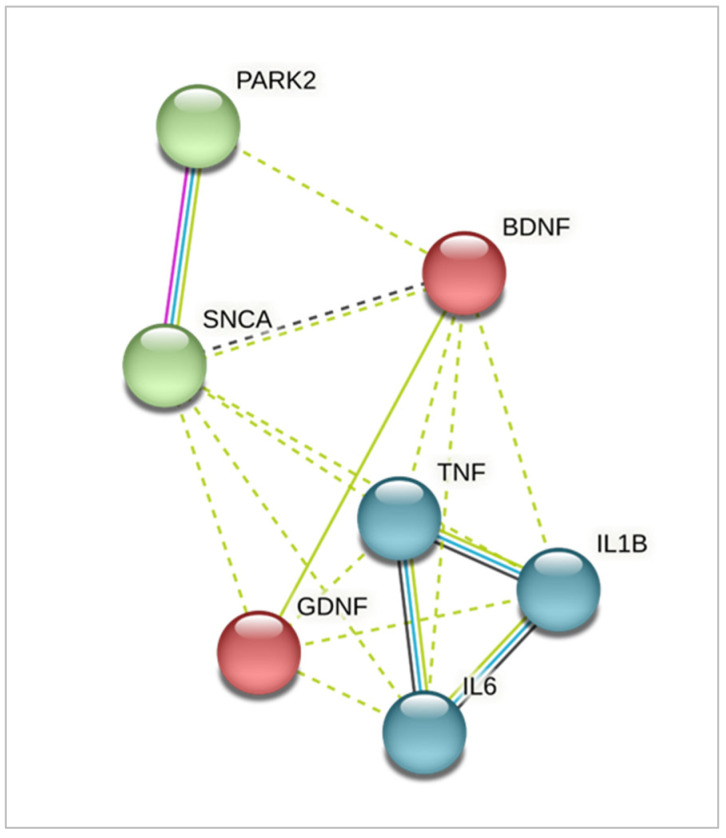
Interaction network of the genes studied using STRING (Search Tool for the Retrieval of Interacting Genes/Proteins). Identified clusters are shown in red, green, and blue. Solid and the dotted lines indicate connection within the same cluster and separate clusters, respectively. Line color indicates the type of interaction evidence: blue—from curated databases; pink—experimentally determined; green—from text mining; black—co-expression.

**Figure 4 ijms-24-02000-f004:**
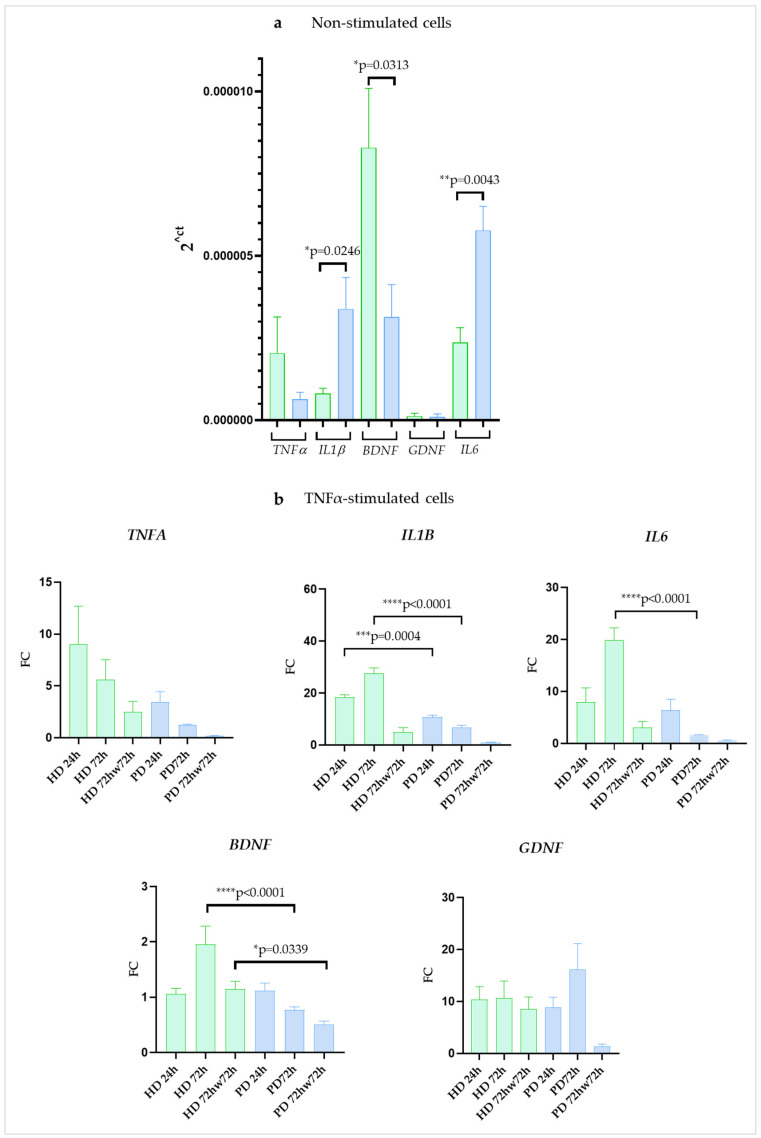
Gene expression in glial cultures derived from iPSCs of HD and patients with *PARK2*-associated PD in steady state and under inflammatory conditions. (**a**), non-stimulated cells; (**b**), cells stimulated with TNFα. Green columns correspond to HD glial cultures, blue—to PD glial cultures. The gene expression was estimated using qPCR (n = 3). Fold change (FC) of expression was calculated relative to mean value in non-stimulated cells at each time point. The data are shown as mean ± SEM (n = 3). Unpaired two-tailed t-test (for non-stimulated cells) and two-way ANOVA with multiple comparisons corrected with Bonferroni test (for stimulated cells) were used for statistical processing. * *p* < 0.05, ** *p* < 0.005, *** *p* < 0.0005, **** *p* < 0.0001.

**Figure 5 ijms-24-02000-f005:**
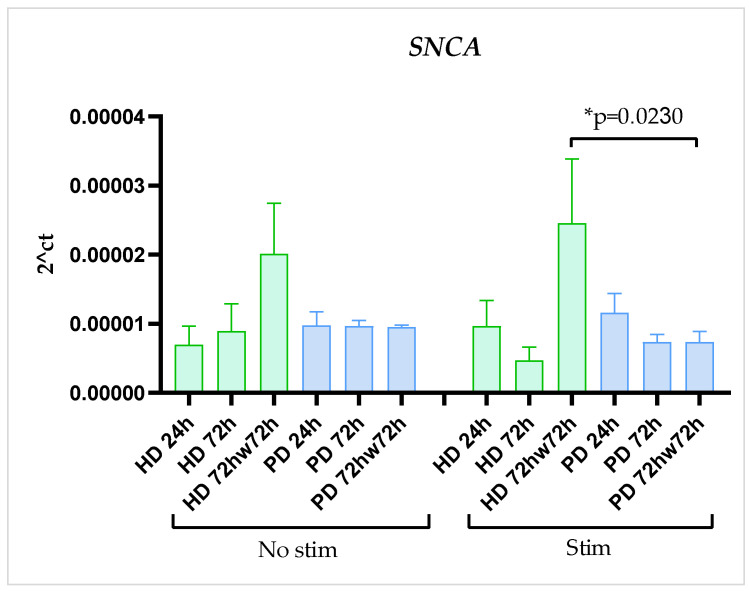
SNCA expression in glial cultures derived from iPSCs of HD and patients with *PARK2*-associated PD in steady state and under inflammatory conditions at different time points of cultivation. Green columns correspond to HD glial cultures, blue—to PD glial cultures. Gene expression was estimated using qPCR (n = 3). The data are shown as mean ± SEM. Two-way ANOVA with multiple comparisons corrected with Bonferroni test was used for statistical processing. * *p* < 0.05.

**Figure 6 ijms-24-02000-f006:**
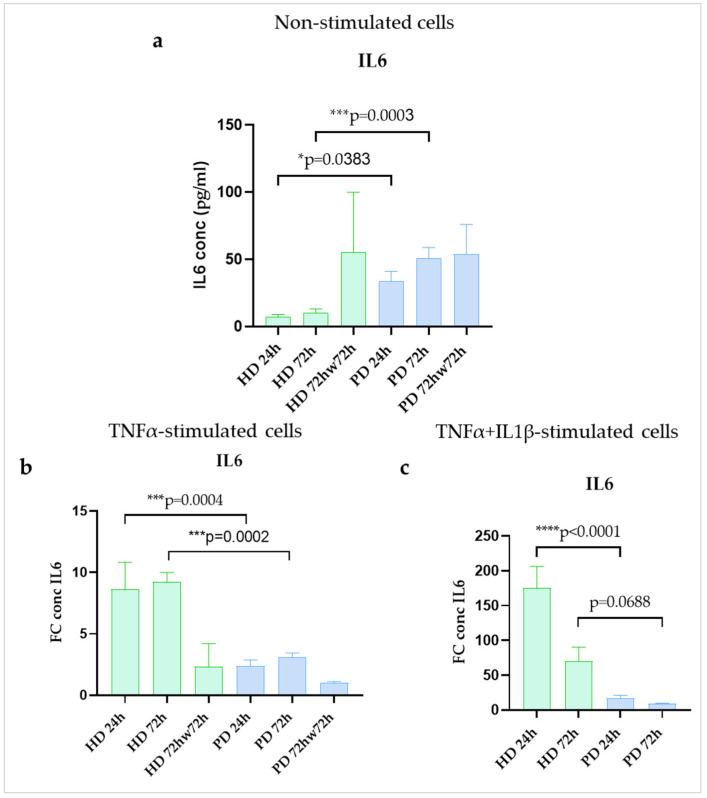
IL6 production in glial cultures derived from iPSCs of HD and patients with *PARK2*-associated PD in steady state (**a**) and under inflammatory conditions (**b**,**c**). Green columns correspond to HD glial cultures, blue—to PD cultures. a. non-stimulated cells. Supernatants were collected at different time points of cultivation. IL6 concentrations were estimated using an ELISA assay (n = 3) and recalculated per 1 × 10^3^ cells (pg/mL). The fold change (FC) of the IL6 concentration was calculated relative to mean value in non-stimulated cells at each time point. The data are shown as mean ± SEM. Two-way ANOVA with multiple comparisons corrected with Bonferroni test (for stimulated cells) were used for statistical processing. * *p* < 0.05, *** *p* < 0.0005, **** *p* < 0.0001.

**Figure 7 ijms-24-02000-f007:**
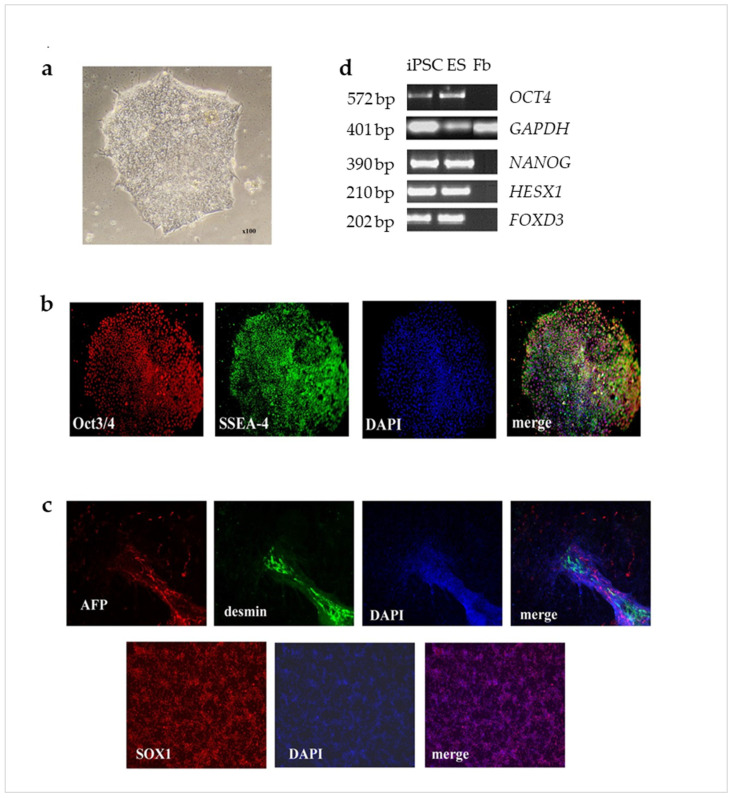
Immunocytochemical staining of obtained iPSC cells. (**a**), typical normal morphology for human pluripotent stem cell; (**b**), the iPSC line demonstrated positive immunofluorescent staining for key transcriptional factor OCT3/4 (red) and surface marker SSEA4 (green); nuclear staining DAPI (blue). ×100; (**c**), the pluripotency was proved by the formation of embryonic bodies, which contained tissues from three germ layers: ectodermal (SOX1—red), mesodermal (desmin—green) and endodermal (AFP—red); nuclear staining (DAPI—blue). ×100; (**d**), the qPCR results demonstrating that *OCT4*, *NANOG*, *HESX1*, and *FOXD3* were expressed in iPSC at a similar level compared to ES (positive control) and were not expressed in fibroblasts (negative control).

**Table 1 ijms-24-02000-t001:** Cell lines description.

Name of Glial Cell Lines	PD Patients and HD	Genotype	Cell Line Symbol
HD1	Healthy male, 60 years	normal	IPSRG2L
HD2	Healthy female, 18 years	normal	IPSHD1.1S
PD1	Male with PD, the disease onset—40 years, biopsy—54 years	*het EX2 dup PARK2*	IPSPDPS8
PD2	Male with PD, the disease onset—38 years, biopsy—40 years	*het EX2 del* PARK2	IPSPDPS2d

**Table 2 ijms-24-02000-t002:** Antibodies used in the study.

Marker	Antibodies	Dilution	Company, Cat #
Pluripotency	Rabbit anti-Oct4	1:200	Abcam, # ab13742
	Mouse anti-SSEA4	1:100	Abcam, # ab16287
Differentiation	Mouse anti-AFP	1:200	Abcam, # ab 3980
	Rabbit anti-desmin	1:200	Abcam, # 15200
	Rabbit anti-Sox1	1:300	Abcam, # 87775
	Rabbit anti-S100	1:2	Agilent Dako, # GA50461-2
Secondary antibodies	Goat anti-Rabbit IgG (H + L), AF546	1:1000	TermoFisher, # A11010
	Goat anti-Mouse IgG (H + L), AF488	1:1000	TermoFisher, # A11008

**Table 3 ijms-24-02000-t003:** The sequences of primers used in the study.

Gene	Forward	Reverse	t^o^ Annealing
*OCT4*	CGACCATCTGCCGCTTTGAG	CCCCCTGTCCCCCATTCCTA	69 °C
*SOX2*	TCCTGATTCCAGTTTGCCTC	GCTTAGCCTCGTCGATGAAC	69 °C
*NANOG*	CAGCCCTGATTCTTCCACCAGTCCC	TGGAAGGTTCCCAGTCGGGTTCACC	69 °C
*FOXD3*	CAAGCCCAAGAACAGCCTAGTGAA	TGACGAAGCAGTCGTTGAGTGAGA	63 °C
*HESH1*	ACCTGCAGCTCATCAGGGAAAGAT	AAAGCAGTTCTTGGTCTCGGCCT	60 °C
*GAPDH*	GAAGGTGAAGGTCGGAGTCA	TTCACACCCATGACGAACAT	60 °C
*TNFA*	CTCCAGGCGGTGCTTGTT	AGGCTTGTCACTCGGGGTT	60 °C
*IL1B*	GCTCGCCAGTGAAATGATGG	GTCCTGGAAGGAGCACTTCAT	60 °C
*IL6*	CCTTCCAAAGATGGCTGAAA	CAGGGGTGGTTATTGCATCT	60 °C
*BDNF*	ATTGGCTGGCGATTCATAAG	GTTTCCCTTCTGGTCATGGA	60 °C
*GDNF*	TGGCTCTGGGCTATGAAACC	ATGCCTGCCCTACTTTGTCA	60 °C
*SNCA*	AGTGACAAATGTTGGAGGAG	GCTTCAGGTTCGTAGTCTTG	60 °C
18S	CGGCTACCACATCCAAGGAA	GCTGGAATTACCGCGGCT	60 °C

## Data Availability

Not applicable.

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
