# Peer review of "Glial Cultures Differentiated from iPSCs of Patients with PARK2-Associated Parkinson’s Disease Demonstrate a Pro-Inflammatory Shift and Reduced Response to TNFα Stimulation"

_ijms, 2023, doi:10.3390/ijms24032000_

Round 1

Reviewer 1 Report

The article by Gerasimova et al., titled “Glial cultures differentiated from iPSCs of patients with PARK2-associated Parkinson's disease demonstrate a pro-inflammatory shift and reduced response to TNFα stimulation” contains interesting perspectives of comparative analysis of TNFα dependent pro-inflammatory shift in the iPSC derived glial cell cultures obtained from healthy donors and PD patients. The authors have shown in-depth knowledge of the subject. It is a well-planned study and the results were presented clearly which supports the conclusion. I have no major concerns.

Author Response

We would like to take this opportunity to thank you for the effort and expertise that you contributed towards reviewing our manuscript. Your positive feedback is greatly appreciated by the whole team.

Please do not hesitate to let us know if you have any further questions.

Reviewer 2 Report

Importantly, the Authors demonstrated a pro-inflammatory shift, suppression of the neuroprotective genetic program in PARK2-deficient glia when compared to glial cells of HDs. The Introduction and bibliography are accurate and updated. The Results can be improved.

In my opinion, this paper is acceptable in IJMS with minor issues as it follows:

-In Figure 2 p values obtained by a multiple comparison between groups (<2) should be inserted. 

-In my opinion, ANOVA test in the place of T-test, should be applied in order to compare different groups (>2) including HD or PD glial cultures.

Author Response

Thank you for the effort and expertise that you contributed towards reviewing our manuscript.

As per your suggestion, we have inserted p-values obtained by a multiple comparison between groups into Figure 2 and used a two-way ANOVA test in place of t-test to compare HD and PD glial cultures. Please find the revised manuscript attached. For clarity, all revisions are highlighted in blue and Figures updated accordingly.

Your insightful comments have allowed us to greatly improve the quality of our Results section. Please do not hesitate to let us know if you have any further questions.

Reviewer 3 Report

Glial cultures differentiated from iPSCs of patients with PARK2-associated Parkinson's disease demonstrate a pro-inflammatory shift and reduced response to TNFα stimulation.

This is an interesting study that assesses the role of glia in the development of Parkinson's disease. To highlight its development in human cell cultures

It seems evident that PARK2 decreases glial capacity in terms of its response to inflammatory processes in relation to healthy patients. However, I propose extending the study to mixed cultures of astrocytes and neurons to assess whether human glial cells actually exert different protective functions on neurons, not only in inflammatory processes, but also in oxidants and others.

I propose that these experiments be carried out or that these aspects of the interaction between glia and neurons in Parkinson's disease be detailed.

Thanks

Author Response

Thank you for taking the necessary time and effort to review our manuscript.

As per your recommendation, we have outlined additional protective functions of astrocytes, not related to inflammatory processes, both in healthy donors and PD patients. Please refer to lines 50-63 in the revised manuscript attached. For clarity, all revisions are highlighted in blue.

We would also like to thank you for your suggestion of expanding our research to cover additional mechanisms of influence of astrocytes on neurons. Although experiments on co-cultivation of neurons and glial cells were not part of the scope of this study, we do plan on carrying out such experiments in the future to form the basis for our next article. Thank you for sharing this idea with us.

Please do not hesitate to let us know if you have any further questions.